# Identifying and characterizing superspreaders of low-credibility content on Twitter

**Matthew R. DeVerna**[1]*, **Rachith Aiyappa**[1], **Diogo Pacheco**[1,2], **John Bryden**[1], **Filippo Menczer**[1]¤

1 Observatory on Social Media, Indiana University, Bloomington, Indiana, United States of America,
2 Department of Computer Science, University of Exeter, Exeter, United Kingdom

¤ Current address: Luddy Center for Artificial Intelligence, Bloomington, Indiana, United States of America
* mdeverna@iu.edu

## Abstract

The world's digital information ecosystem continues to struggle with the spread of misinformation. Prior work has suggested that users who consistently disseminate a disproportionate amount of low-credibility content—so-called superspreaders—are at the center of this problem. We quantitatively confirm this hypothesis and introduce simple metrics to predict the top superspreaders several months into the future. We then conduct a qualitative review to characterize the most prolific superspreaders and analyze their sharing behaviors. Superspreaders include pundits with large followings, low-credibility media outlets, personal accounts affiliated with those media outlets, and a range of influencers. They are primarily political in nature and use more toxic language than the typical user sharing misinformation. We also find concerning evidence that suggests Twitter may be overlooking prominent superspreaders. We hope this work will further public understanding of bad actors and promote steps to mitigate their negative impacts on healthy digital discourse.

**Data Availability Statement:** The code and data for this study can be found at: github.com/osome-iu/low-cred-superspreaders. In compliance with the terms of our contract with Twitter to access the Decahose data, we are only permitted to release

## Introduction

Misinformation impacts society in detrimental ways, from sowing distrust in democratic institutions to harming public health. The peaceful transition of power in the United States was threatened on January 6[th], 2021 when conspiracy theories about the presidential election being "stolen" fueled violent unrest at the U.S. Capitol [1]. During the COVID-19 pandemic, an abundance of health-related misinformation spread online [2, 3], ultimately driving the U.S. Surgeon General to warn Americans about the threat of health misinformation [4]. The public confusion created by this content led the World Health Organization to collaborate with major social media platforms and tech companies across the world in an attempt to mitigate its spread [5].

Recent research suggests that *superspreaders* of misinformation—users who consistently disseminate a disproportionately large amount of low-credibility content—may be at the center of this problem [2, 6–11]. In the political domain, one study investigated the impact of misinformation on the 2016 U.S. election and found that 0.1% of Twitter users were responsible for sharing approximately 80% of the misinformation [6]. Social bots also played a

the tweet IDs. These data can be reconstructed using the X API (https://developer.twitter.com/en/docs/twitter-api) which, unfortunately, now requires a paid subscription. However, other collected data is available.

**Funding:** This work was supported by the John S. and James L. Knight Foundation, Craig Newmark Philanthropies, and the National Science Foundation (grant ACI-1548562). The funders had no role in study design, data collection and analysis, decision to publish, or preparation of the manuscript.

**Competing interests:** The authors declare no competing interests.

disproportionate role in spreading content from low-credibility sources [12]. The Election Integrity Partnership (a consortium of academic and industry experts) reported that during the 2020 presidential election, a small group of "repeat spreaders" aggressively pushed false election claims across various social media platforms for political gain [8, 9].

In the health domain, analysis of the prevalence of low-credibility content related to the COVID-19 "infodemic" on Facebook and Twitter showed that superspreaders on both of these platforms were popular pages and accounts that had been verified by the platforms [2]. In 2021, the Center for Countering Digital Hate reported that just 12 accounts—the so-called "disinformation dozen"—were responsible for almost two-thirds of anti-vaccine content circulating on social media [10, 11]. This is concerning because eroding the public's trust in vaccines can be especially dangerous during a pandemic [13] and evidence suggests that increased exposure to vaccine-related misinformation may reduce one's willingness to get vaccinated [14, 15].

Despite the growing evidence that superspreaders play a crucial role in the spread of misinformation, we lack a systematic understanding of who these superspreader accounts are and how they behave. This gap may be partially due to the fact that there is no agreed-upon method to identify such users; in the studies cited above, superspreaders were identified based on different definitions and methods.

In this paper, we tackle this gap by providing a coherent characterization of superspreaders of low-credibility content on Twitter. In particular, we address two research questions. First, **(RQ1) can superspreaders of low-credibility content be reliably identified?** To be useful, any method for measuring the degree to which an online account is a superspreader of such content should be accurate and predictive. Here we focus on simple approaches utilizing data that are widely available across platforms. More complex methods may require detailed information about the structure of the entire social network, which is typically unavailable.

Mitigating the negative impact of superspreaders of low-credibility content additionally requires a deeper understanding of these users, leading to our second research question: **(RQ2) who are the superspreaders, i.e., what types of users make up most superspreader accounts and how do they behave?** A better understanding of the origins of misinformation is an important step toward decreasing its amplification and reach [16].

To answer our first research question, we begin by collecting 10 months of Twitter data and defining "superspreaders" as accounts that introduce low-credibility content, which then disseminates widely. Operationally, we define low-credibility content as content originally published by low-credibility, or untrustworthy sources. With this definition, we evaluate various platform-agnostic metrics to predict which users will continue to be superspreaders after being identified. The labeling of sources and the metrics are detailed in Methods, below. We do this by ranking accounts in an initial time period with each metric and then comparing how well these rankings predict a user will be a superspreader in a subsequent period. We also compare all metrics to an optimal performance based on data from the evaluation period. The metrics considered are based on *Bot Score* (likelihood that an account is automated, calculated utilizing BotometerLite [17]), *Popularity* (number of followers), *Influence* (number of retweets of low-credibility content earned during the initial period), and *h-index*, repurposing a metric initially proposed to study scholarly impact [18]. We find that the *h*-index and Influence metrics outperform other metrics and achieve near-optimal accuracy in predicting the top superspreaders.

After validating the *h*-index and Influence metrics, we address our second research question by conducting a qualitative review of the worst superspreaders. Behavioral statistics and relevant user characteristics are analyzed as well; e.g., whether accounts are verified or suspended. This allows us to provide a qualitative description of the superspreader accounts we

identify. 52% of superspreaders on Twitter are political in nature. We also find accounts of pundits with large followings, low-credibility media outlets, personal accounts affiliated with those media outlets, and a range of nano-influencers—accounts with around 14 thousand followers. Additionally, we learn that superspreaders use toxic language significantly more often than the typical user sharing low-credibility content. Finally, we examine the relationships between suspension, verified status, and popularity of superspreaders. This analysis suggests that Twitter may overlook verified superspreaders with very large followings.

## Related work

There is a great deal of literature on the identification of influential nodes within a network [19]. While some of this work is not directly related to the social media space, it offers some guidance about how nodes—in our case, accounts—interact within an information diffusion network. Given that the dynamics of diffusion are hard to infer, this work often takes a structural and/or a topical approach.

Structural approaches focus on extracting information about potentially influential users from the topology of social connections in a network [20–24]. A classic example is PageRank, an algorithm that counts the number and quality of connections to determine a node's importance [21]. Several authors have found that the $k$-core decomposition algorithm [25, 26] outperforms other node centrality measures in identifying the most effective spreaders within a social network [23, 24]. This algorithm recursively identifies nodes that are centrally located within a network. Unfortunately, this method is unable to differentiate between individuals in the network's core.

Topical approaches take into account network structure and also consider the content being shared [27, 28]. For example, Topic Sensitive PageRank [27] calculates topic-specific PageRank scores. Another way to extend PageRank is to bias the random walk through a topic-specific relationship network [28].

Given the ample evidence of manipulation within social media information ecosystems [7, 12, 29, 30], it is important to investigate whether the results mentioned above generalize to misinformation diffusion on social media platforms. Simple heuristics like degree centrality (i.e., the number of connections of a node) perform comparably to more expensive algorithms when seeking to identify superspreaders [31]. These results, though encouraging, rely on model-based simulations and decade-old data. More recent work has proposed methods for identifying fake news spreaders and influential actors within disinformation networks that rely on deep neural networks and other machine learning algorithms [32, 33]. These methods, however, are hard to interpret. Another approach is to pinpoint the smallest possible set of influential accounts by applying optimal percolation to the diffusion network [34–36]. This is similar to the network dismantling method used in our evaluation of different metrics to rank superspreaders.

Here we evaluate simple metrics inspired by the literature [23, 31] that can be applied easily to most social media platforms, and address the gaps related to the misinformation space. We consider a Twitter user's degree within the social (follower) network as well as the diffusion (retweet) network of low-credibility content. We also apply the $h$-index [18]—previously proposed as a measure of node influence [37]—in a novel way to the realm of misinformation. Finally, we consider a bot score metric [17] that has been shown to capture the role of potential platform manipulation by inauthentic accounts [12].

This paper addresses two research questions. To address RQ1, we collect a dataset of low-credibility content spreading on Twitter. We then compare different metrics for identifying superspreaders of this content and provide details about these metrics as well as optimal

performance. A dismantling analysis [7, 12] is utilized to quantify how much *future low-credibility content* each user is responsible for spreading. To characterize the identified superspreaders (RQ2), we focus on the worst-offending accounts. We manually classify them into different categories and then describe their behavior.

## Methods

### Low-credibility content diffusion

We begin this analysis by building a low-credibility content diffusion dataset from which we can identify problematic users. To identify this content, we rely on the *Iffy+* list [38] of 738 low-credibility sources compiled by professional fact-checkers—an approach widely adopted in the literature [2, 6, 12, 35, 39]. This approach is scalable, but has the limitation that some individual articles from a low-credibility source might be accurate, and some individual articles from a high-credibility source might be inaccurate.

Tweets are gathered from a historical collection based on Twitter's Decahose Application Programming Interface (API) [40]. The Decahose provides a 10% sample of all public tweets. We collect tweets over a ten-month period (Jan. 2020–Oct. 2020). We refer to the first two months (Jan–Feb) as the *observation* period and the remaining eight months as the *evaluation* period. From this sample, we extract all tweets that link to at least one source in our list of low-credibility sources. This process returns a total of 2,397,388 tweets sent by 448,103 unique users.

### Metrics

Let us define several metrics that can be used to rank users in an attempt to identify superspreaders of low-credibility content.

**Popularity.** Intuitively, the more followers you have on Twitter, the more your posts are likely to be seen and reposted. As a simple measure of popularity, we can use an account's number of followers, even though it does not fully capture its influence [41]. Specifically, let us define Popularity as the mean number of Twitter followers an account had during the observation period. We extracted the numbers of followers from the metadata in our collection of tweets.

**Influence.** Various measures of social media influence have been proposed [41]. One that is directly related to spreading low-credibility content can be derived from reshares of posts that link to untrustworthy sources. We compute the Influence $\mathcal{I}$ of account $i$ by summing the number of retweets of all posts they originated that link to low-credibility sources during our observation period. This is formally expressed as $\mathcal{I}_i = \sum_{t \in \mathcal{T}_i} \rho_t$, where $\rho_t$ denotes the number of retweets of post $t$, and $\mathcal{T}_i$ is the set of all observed posts by account $i$ that link to low-credibility content. One could also consider quoted tweets, however we focus on retweets because they are commonly treated as endorsements; quoted tweets can indicate other intent such as criticism.

**Bot score.** Some research has reported that social bots can play an important role in the spread of untrustworthy content [12]. Therefore, we adopt a Bot Score metric that represents the likelihood of an account being automated [42]. A user's Bot Score is given by the popular machine learning tool BotometerLite [43], which returns a score ranging from zero to one, with one representing a high likelihood that an account is a bot. Machine learning models are imperfect but enable the analysis of significantly larger datasets. BotometerLite is selected for its high accuracy, which will minimize error, and its reliance only on user metadata from the Twitter V1 API [17]. This allows us to analyze the user objects within our historical data,

calculating the likelihood that a user was a bot *at the time of observation*; as opposed to relying on other popular tools that query an account's most recent activity at the time of estimation [44]. Since we obtain a score from the user object in each tweet, we set user *i*'s Bot Score equal to the mean score across all tweets by *i* in the observation period.

**h-index.**   To quantify an account's consistent impact on the spread of content from low-credibility sources, we repurpose the *h*-index, which was originally developed to measure the influence of scholars [18]. The *h*-index of a scholar is defined as the maximum value of *h* such that they have at least *h* papers, each with at least *h* citations. Similarly, in the context of social media, we define $h(i)$ of user *i* as the maximum value of *h* such that user *i* has created at least *h* posts *linking to low-credibility sources*, each of which has been reshared at least *h* times by other users.

We apply this metric to the Twitter context and adopt the most common metric on this platform for resharing content, the retweet count. As a result, a Twitter user *i* with $h = 100$ means that the user has posted at least 100 tweets linking to low-credibility sources, each of which has been retweeted at least 100 times.

Unlike common measures of influence, such as the retweet count or the number of followers, this repurposing of the *h*-index focuses on problematic repeat-offenders by capturing the *consistency* with which a user shares low-credibility content [45]. For example, a user *i* who posts only one untrustworthy tweet that garners a large number of retweets earns $h = 1$, regardless of the virality of that individual tweet.

## Accounting for future low-credibility content

This work seeks to predict which Twitter accounts will be superspreaders of untrustworthy content in the future. To this end, we identify accounts in the observation period and then quantify how much low-credibility content they spread during the evaluation period. We construct a retweet network with the data from each period. The observation network (Jan–Feb 2020) and the evaluation network (Mar–Oct 2020) involve approximately 131 thousand and 394 thousand users, respectively. In each network, nodes represent accounts and directed edges represent retweets pointing from the original poster to the retweeter. Each edge $(i, j)$ is weighted by the total number $w_{ij}$ of times any of *i*'s posts linking to low-credibility content are retweeted by *j*.

We create four separate rankings of the 47,012 users that created at least one post linking to low-credibility content during the observation period based on each of the metrics defined above: *h*-index, Popularity, Influence, and Bot Score.

For each ranking, we employ a network dismantling procedure [7, 12] wherein accounts are removed one by one in order of ascending rank from the retweet network. As we remove account *i* from the network, we also remove all retweets of posts linking to low-credibility content originated by *i*, i.e., the outgoing edges from *i*. We can calculate the proportion of untrustworthy content removed from the network with the removal of account *i* as

$$M_i = \frac{\sum_j w_{ij}}{\sum_{kj} w_{kj}},$$ (1)

where the denominator represents the sum of all edge weights prior to beginning the dismantling process. This quantifies how much low-credibility content each user is responsible for during the evaluation period.

Note that Twitter's metadata links all retweets of a tweet to the original poster. Therefore, the value $M_i$ for each account *i* is the same across all ranking algorithms. The performance of a metric depends only on the order in which the nodes are removed, determined by the metric-

based ranking. We compare how quickly the metrics remove low-credibility content from the network relative to one another. Metrics that remove this content most quickly are considered the best ones for identifying superspreaders. This is because they rank the accounts responsible for disseminating the largest proportion of low-credibility content at the top.

We also compare each ranking to the optimal ranking for our dismantling-based evaluation. This optimal strategy is obtained by ranking candidate superspreaders according to descending values of $M$, where $M$ is calculated by using the evaluation period instead of the observation period. That is, the account with the largest $M$ value is removed first, followed by the one with the second largest $M$, and so on, until all users have been removed. Note that this optimal ranking is only possible using information from the future evaluation period as an oracle. It serves as an upper bound on the performance that can be expected from any ranking metric.

## Account classification and description

The top superspreader accounts according to the rankings described above are classified into one of the 16 different categories detailed in Table 1. We adopted and slightly altered a classification scheme from a previous study [45]. Health-related and COVID-19-specific categories, i.e., "public health official," "medical professional," and "epidemiologist," were removed. A "media affiliated" category was added to capture accounts that might have some affiliation with low-credibility sources, as seen in previous research [2]. This classification scheme takes into account different types of journalists as well as other influential individuals and entities, such as politicians, media outlets, religious leaders, and organizations. Additionally, accounts in certain categories ("elected official" and "political") are annotated with their political affiliation: "right" (conservative) or "left" (liberal). The same is done for hyperpartisan accounts in certain other categories, such as media and journalists.

Two authors independently annotated each account. In cases of disagreement, two additional authors followed the same process. The category and political affiliation of these accounts were then derived from the majority classification (three of the four annotators). Accounts for which the disagreement could not be resolved were excluded.

**Table 1. Classification scheme utilized during the process of manually annotating superspreader accounts.** An account's political affiliation was recorded if an annotator classified that account as political. The same was done for hyperpartisan accounts in certain other categories, such as media and journalists.

| Classification | Examples | Political Affiliation |
|---|---|---|
| Elected official | Mayors, governors, senators | Recorded |
| Public service | City offices, public departments | |
| Media outlet | News outlets, TV news channels | If hyperpartisan |
| Journalist (hard news) | Investigative journalists, public health and economics reporters | If hyperpartisan |
| Journalist (soft news) | Sports and entertainment reporters | If hyperpartisan |
| Journalist (broadcast news) | TV anchors, radio show hosts | If hyperpartisan |
| Journalist (new media) | Twitch streamers, podcast hosts | If hyperpartisan |
| Media affiliated | Editors, high-level employees, owners of media outlets | If hyperpartisan |
| Public intellectual | Academic researchers, mainstream opinion columnists | |
| Political | Activists, campaign staffers, political personalities, political pundits, anonymous hyperpartisan accounts | Recorded |
| Entertainer | Musicians, comedians, social media personalities | |
| Sports related | Baseball players, sports managers | |
| Religious leader | Priests, rabbis, churches | |
| Organization | Organizations not classified elsewhere | |
| Other | Accounts not classified elsewhere. Primarily personal accounts of non-public figures with moderate followings | |
| Deactivated/suspended | Accounts deactivated/suspended at the time of annotation | |

## Source-sharing behavior

We investigate the typical behavior of a top superspreader account with respect to sharing low-credibility sources, relative to their general source-sharing behavior. Specifically, for a given account, we calculate the ratio $r_m = \frac{|\mathcal{T}_i|}{|\mathcal{P}_i|}$, where $|\mathcal{T}_i|$ represents the total count of user $i$'s posts that link to low-credibility sources and $|\mathcal{P}_i|$ is the count of all posts by user $i$ that link to any source during the observed period. This also allows us to better understand the proportion $1 - r_m$ of non-low-credibility sources that would be lost if the account were removed. This type of content may originate from trusted sources and is assumed to be harmless. An ideal method would identify users that consistently share high-impact untrustworthy content *and* a minimal proportion of harmless content.

To calculate $r_m$, we first download *all* tweets sent by the identified superspreaders during a three-month period (Jan 1, 2020–April 1, 2020). We were able to gather tweets from 123 superspreader accounts that were still active. We then extract all links from the metadata of these tweets. We expand links that are processed by a link-shortening service (e.g., bit.ly) prior to being posted on Twitter. Sources are obtained by extracting the top-level domains from the links. Low-credibility sources are identified by matching domains to the *Iffy+* list described earlier. Finally, we calculate the proportion $r_m$ for all superspreaders. The inability to calculate $r_m$ for inactive accounts might introduce bias in this measurement.

## Language toxicity

We wish to investigate the content of superspreader posts beyond source-sharing behaviors to understand if they are taking part in respectful discourse or increasing the levels of abusive language in public discussion. We utilize the Google Jigsaw Perspective API [46] to estimate the probability of each tweet in the 10-month dataset being toxic. The API defines toxic language as rude, disrespectful, or unreasonable comments that are likely to make users disengage from an online interaction. We then calculate the toxicity of an account by averaging the score across all of their original tweets. We only consider English-language tweets; five superspreaders tweeting exclusively in other languages are excluded.

While recognizing the model's "black box" nature as a limitation, we still embrace its adoption, aligning with prevailing practices in social media research. This approach ensures our work's comparability with other pertinent studies [47].

# Results

## Dismantling analysis

After ranking accounts in the observation period based on the investigated metrics (*h*-index, Popularity, Influence, and Bot Score), we conduct a dismantling analysis to understand the efficacy of each one (see Methods for details). The results of this analysis are show in Fig 1 (top).

Bot Score performs the worst: even after more than 2,000 accounts are removed from the network, most of the low-credibility content still remains in the network. This suggests that bots infrequently *originate* this content on Twitter. Instead, as previous research suggests, bots may increase views through retweets and expose accounts with many followers to low-credibility content, in hopes of having them reshare it for greater impact [12].

We also observe in Fig 1 (top) that while Popularity performs substantially better than Bot Score, it fails to rank the most problematic spreaders at the top; upon removing the top 10 users, almost no low-credibility content is removed from the network. In contrast, the *h*-index and Influence metrics place superspreaders at the top of their rankings and the dismantling

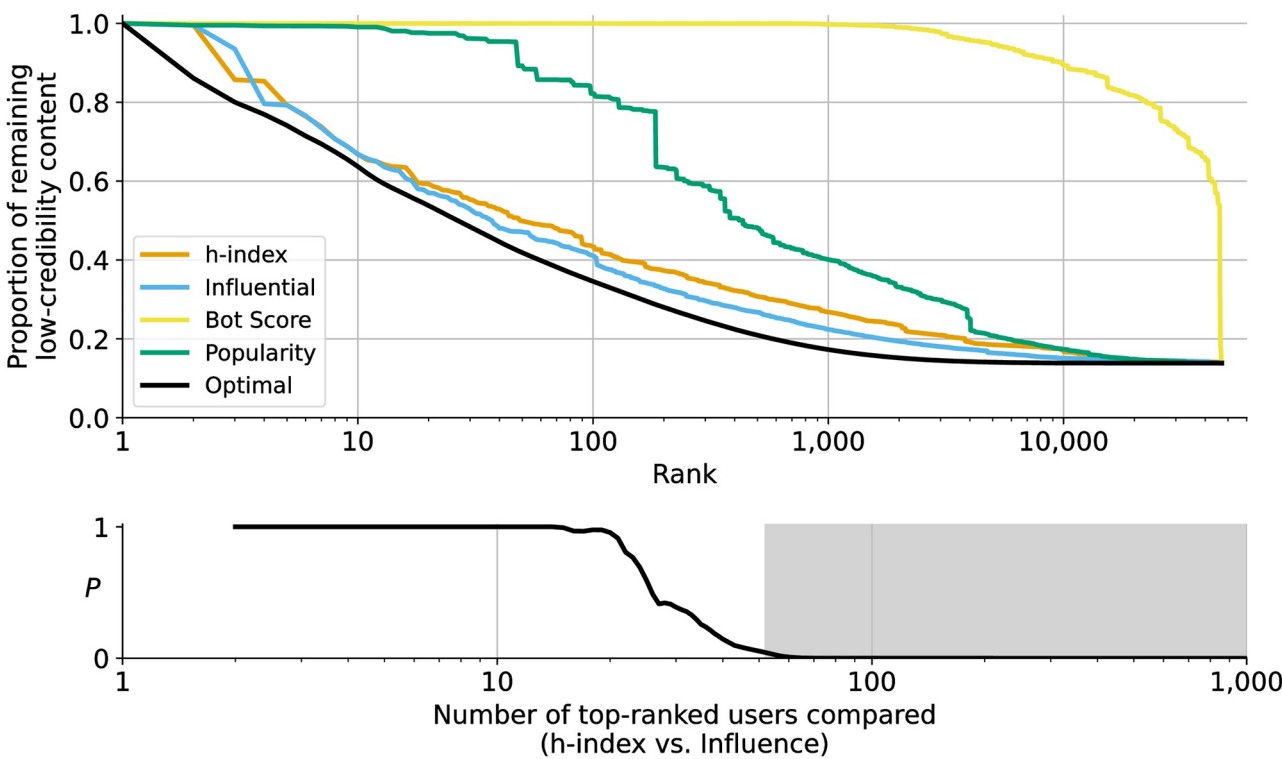

**Fig 1.** *Top*: The effect of removing accounts that created low-credibility posts during January and February 2020 (observation period) on the proportion of untrustworthy content present during the following eight months (evaluation period). Nodes (accounts) are removed one by one from a retweet network in order of ascending rank, based on the metrics indicated in the legend. The remaining proportion of retweets of low-credibility posts is plotted versus the number of nodes removed. The lowest value for all curves is not zero, reflecting the fact that approximately 13% of the low-credibility retweets in the evaluation network are by accounts who did not create low-credibility posts during the observation period. *Bottom*: Likelihood that the difference between the performance of *h*-index and Influence happened by random chance. The most prolific superspreaders according to these two metrics remove a similar amount of low-credibility content. To compare them for any given number of removed accounts, we conduct Cramer von Mises two-sample tests with increasingly larger samples and plot each test's *P*-value on the *y*-axis. After removing more than 50 accounts (gray area) the Influence metric performs significantly better ($P < 0.05$). The difference is not significant if fewer accounts are removed.

procedure removes substantial amounts of low-credibility content from the network immediately.

The Popularity metric draws on the structure of the *follower network* and therefore contains valuable information about how low-credibility content might spread. However, the follower network is not a perfect predictor of diffusion networks [23]. The *retweet network* used by the *h*-index and Influence metrics provides a more direct prediction.

Cramer von Mises (CvM) two-sample comparisons show significant differences between the optimal curve and those for *h*-index ($P < 0.001$, $d = 0.61$, 95% CI: [0.02, 0.02]) and Influence metrics ($P < 0.001$, $d = 0.44$, 95% CI: [0.01, 0.01]). All confidence intervals are calculated based on bootstrapping (5,000 resamples). However, the amount of low-credibility content removed using either metric is within 2% of the optimal, on average. In fact, removing the top 10 superspreaders eliminates 34.6% and 34.3% of the low-credibility content based on *h*-index and Influence, respectively (optimal: 38.1%). In other words, 0.003% of the accounts active during the evaluation period posted low-credibility content that received over 34% of all retweets of this content over the eight months following their identification. Removing the top 1,000 superspreaders (0.25% of the accounts who posted during the evaluation period) eliminates 73–78% of the low-credibility content (optimal: 81%). This represents a remarkable concentration of responsibility for the spread of untrustworthy content.

Comparing the performance of $h$-index and Influence to one another across *all* ranked accounts illustrates that ranking by the Influence metric removes significantly more low-credibility content on average (CvM: $P < 0.001$, $d = 0.22$, 95% CI: [0.01, 0.01]). However, it is more useful to compare the performance between these metrics with respect to the highest ranked accounts, since those would be considered as potential superspreaders. Let us again utilize CvM tests to compare the impact of removing samples of top superspreaders of increasing size, up to 1,000 accounts. We first check if the amount of low-credibility content attributed to the top two ranked accounts according to each metric is significantly different, then the top three, and so on, until we have considered the top 1,000 accounts in each group. As shown in Fig 1 (bottom), rankings by $h$-index and Influence are not significantly different when comparing the amount of low-credibility content attributed to the top-ranked accounts. Only after removing accounts ranked 51st or below—who likely would not be categorized as superspreaders—does the performance of these metrics begin to differ significantly (CvM: $P = 0.048$, $d = 0.17$, 95% CI: [−0.03, 0.07]).

Overall, these results suggest that, with respect to our sample, both $h$-index and Influence metrics perform well at identifying superspreaders of low-credibility content. Since removing accounts based on these two metrics yields similar reductions in untrustworthy content, we explore other reasons to prefer one over the other in later sections.

## Describing superspreaders

In this section we characterize superspreaders of low-credibility content in terms of their account type, untrustworthy content sharing behavior, and use of toxic language. We also investigate the relationship between an account's follower count and its verified or suspended status. The top 1% of accounts with $h$-index above zero are selected as superspreaders, yielding 181 accounts, and then an equal number of top-ranked accounts are taken for comparison, based on the Influence metric. We note that other thresholds could be adopted to classify an account as a superspreader. This approach allows us to focus on a large but manageable number of accounts that have large influence within the low-credibility content ecosystem.

**Account classification.** The groups selected by the two metrics overlap, so there are a total of 250 unique accounts. These were manually classified into different categories following the procedure detailed in Methods. After the first round of classifications, two authors agreed on 211 accounts (84.4%, Krippendorf's $\alpha = 0.79$). Of the remaining 39 accounts reviewed by two additional authors, 21 were classified by a majority of annotators and the rest were excluded, yielding 232 classified accounts.

Fig 2 reports the number of superspreader accounts in each category. Over half of the accounts (55.1%) were no longer active at time of analysis. Of these 128 inactive accounts, 111 (86.7%) were reported by Twitter as suspended. The suspended accounts were evenly distributed among the superspreaders identified by $h$-index (47.5%, 86 accounts) and Influence (42.5%, 78 accounts). The remaining 17 inactive accounts were deleted. The high number of suspensions serves as further validation of these metrics: Twitter itself deemed many of the accounts we identified as superspreaders to be problematic.

The accounts still active were classified according to the scheme in Table 1. 52% (54 accounts) fall into the "political" group. These accounts represent users who are clearly political in nature, discussing politics almost exclusively. They consist largely of anonymous hyperpartisan accounts but also high-profile political pundits and strategists. Notably, this group includes the official accounts of both the Democratic and Republican parties (@*TheDemocrats* and @*GOP*), as well as @*DonaldJTrumpJr*, the account of the son and political advisor of then-President Donald Trump.

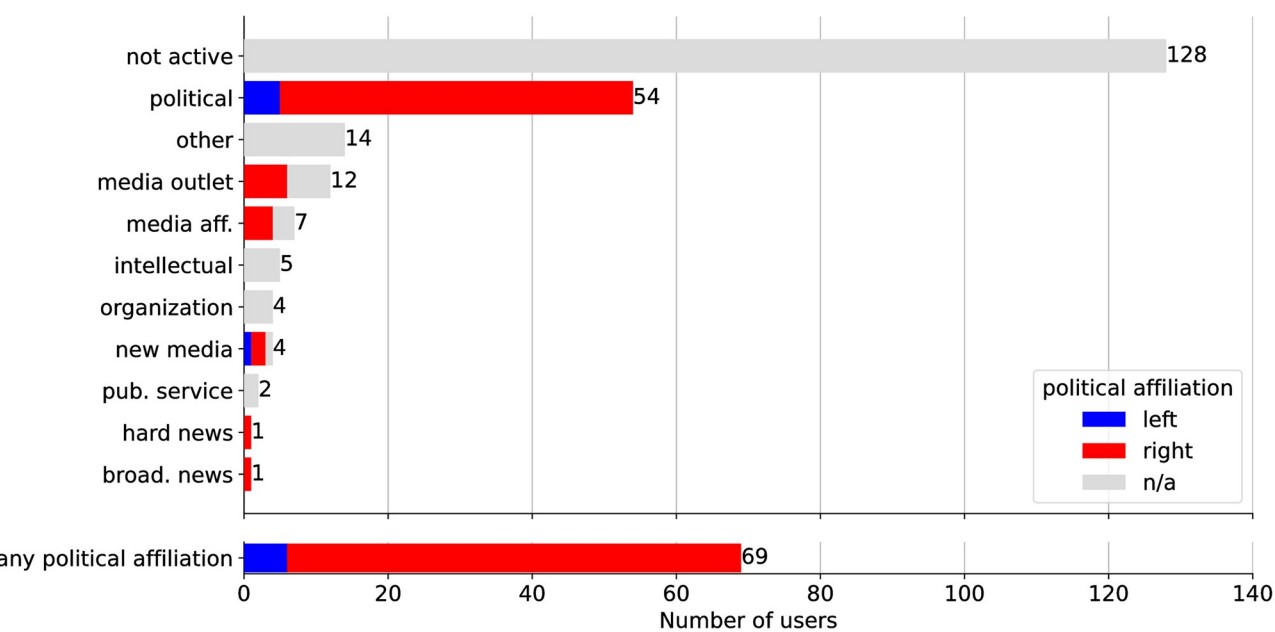

**Fig 2. Classification of superspreader accounts.** A large portion (55.1%) of accounts are no longer active. For each class annotated with political affiliations, colors indicate the ideological split. The last group aggregates all accounts with political affiliations.

The next largest group is the "other" category, making up 14 active accounts (13.4%). This group mostly consists of nano-influencers with a moderate following (median ≈ 14 thousand followers) posting about various topics. A few accounts were classified in this group simply because their tweets were in a different language.

The "media outlet" and "media affiliated" classifications make up the next two largest groups, consisting of 19 active accounts combined (18.3%). Most of the media outlets and media affiliated accounts are associated with low-credibility sources. For example, Breaking911.com is a low-credibility source and the *@Breaking911* account was identified as a superspreader. Other accounts indicate in their profile that they are editors or executives of low-credibility sources.

The remainder of the superspreaders consist of (in order of descending number of accounts) "organizations," "intellectuals," "new media," "public service," "broadcast news," and "hard news" accounts. Notable among these accounts are: the prominent anti-vaccination organization, Children's Health Defense, whose chairman, Robert F. Kennedy Jr., was named as one of the top superspreaders of COVID-19 vaccine disinformation [10, 11, 48]; the self-described "climate science contrarian" Steve Milloy, who was labeled a "pundit for hire" for the oil and tobacco industries [49]; and the popular political pundit, Sean Hannity, who was repeatedly accused of peddling conspiracy theories and misinformation on his show [50–52].

Examining the political ideology of superspreaders, we find that 91% (49 of 54) of the "political" accounts are conservative in nature. Extending this analysis to include other hyper-partisan accounts (i.e., those classified as a different type but still posting hyperpartisan content), 91% of accounts (63 of 69) are categorized as conservative.

Fig 2 also reports political affiliations by superspreader account class. The conservative/liberal imbalance is largely captured within the political accounts group. However, we also see that approximately half of the "media outlet" and "media affiliated" superspreaders consist of hyperpartisan conservative accounts. These results agree with literature that finds an

asymmetric tendency for conservative users to share misinformation online compared to liberal users [6, 53, 54].

**Low-credibility content sharing behavior.** The dismantling analysis focuses on low-credibility content and does not capture the rest of the content shared by an account. This distinction is important because moderation actions, such as algorithmic demotion, suspension, and deplatforming, limit a user's ability to share *any* content. To better understand the full impact of removing superspreaders, we analyze the likelihood that a superspreader shares a low credibility source. We estimate this likelihood using the proportion $r_m$ defined in Methods.

Fig 3 compares the distributions of proportions of low-credibility links shared by the superspreaders identified by the $h$-index and Influence metrics. We see that accounts identified via $h$-index share relatively more low-credibility sources than those identified with the Influence metric; a two-way Mann-Whitney $U$ test confirms that this difference is significant ($p < 0.01$, $d = 0.16$, 95% CI: [−0.04, 0.13]). Specifically, the median proportion of shared sources that are low-credibility for accounts identified by the $h$-index (median = 0.07, mean = 0.22, $n = 84$) is approximately two times larger than for those identified with the Influence metric (median = 0.03, mean = 0.17, $n = 91$). In other words, while removing superspreader accounts based on the two metrics has a similar effect on curbing untrustworthy content, using the $h$-index metric is preferable because it removes less content that is not from low-credibility sources. This result makes sense in light of the fact that the $h$-index prioritizes accounts who share low-credibility sources *consistently*.

**Language toxicity.** Let us now explore the language used by superspreaders. We first compare the distribution of mean toxicity scores for accounts identified by the $h$-index and Influence metric. Toxicity scores are estimated with the Perspective API [46] (see details in Methods).

We find that superspreaders identified by the $h$-index display similar average toxicity (median = 0.18, mean = 0.20, $n = 178$) to those identified with the Influence metric (median = 0.18, mean = 0.20, $n = 179$); a Mann-Whitney U two-way comparison indicates this difference is not significant ($P = 0.61$, $d = 0.01$, 95% CI: [−0.02, 0.02], $n = 245$). Fig 4 shows superspreaders having significantly higher toxicity than all accounts within our dataset ($P < 0.001$, $d = 0.12$, 95% CI: [0.01, 0.03], $n = 149, 481$). However, at the individual level, we observe no significant correlation between toxicity and $h$-index (Spearman $r = 0.03$, $P = 0.67$) or Influence (Spearman $r = 0.08$, $P = 0.26$).

**Account prominence.** Approximately one in five of the superspreader accounts (48 out of 250) have been verified by Twitter. Given such a large proportion of verified accounts, we investigate the relationship between the prominence (verified status, followers, and retweets) and active/suspended status of these accounts.

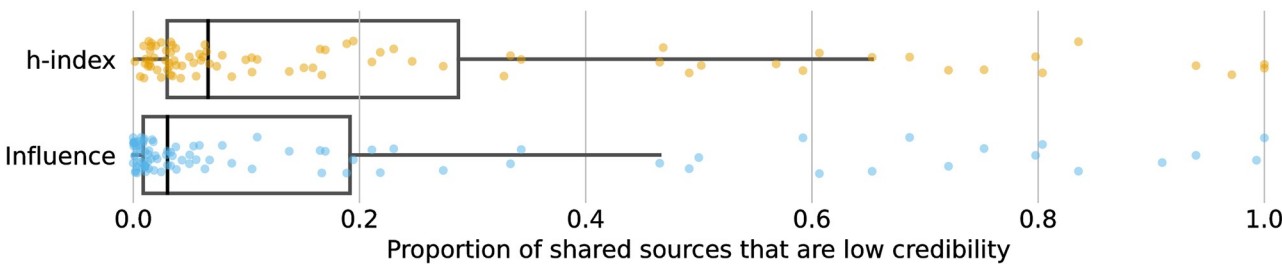

**Fig 3. Low-credibility content sharing behavior of superspreaders (points) as captured by the boxplot distribution of the ratio $r_m$.** Users identified via the $h$-index share a significantly higher ratio of untrustworthy sources than those identified with the Influence metric.

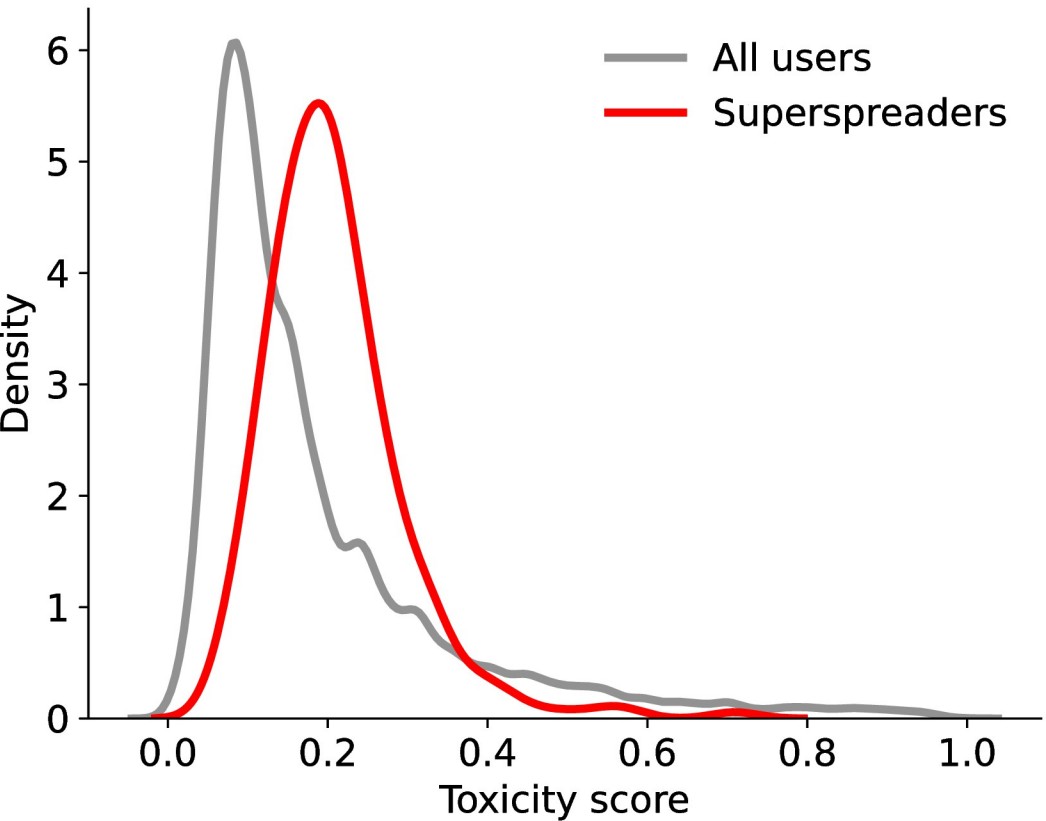

**Fig 4. Distributions of language toxicity scores for superspreaders vs. all accounts in the low-credibility content ecosystem.**

Fig 5 (top) shows that more prominent superspreaders are less likely to be suspended: only 3% of suspended accounts were verified. As shown in Fig 5 (bottom), superspreaders with many (more than 150 thousand) followers are also less likely to have been suspended. A similar pattern is observed using different thresholds for the number of followers.

Additionally, we find a significant correlation between a superspreader's number of followers and the amount of low-credibility content they were responsible for ($M$) during the evaluation period (Spearman $r = 0.42$, $P < 0.001$).

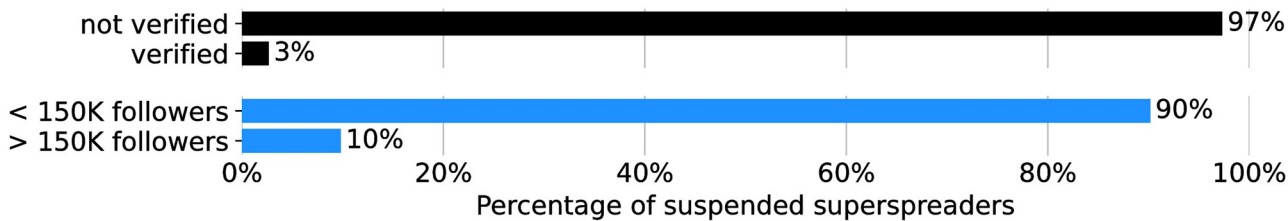

**Fig 5. Relationship between suspension, verified status, and popularity of top 250 superspreaders.** *Top*: Percentage of suspended superspreader accounts that are verified. *Bottom*: Percentage of suspended superspreader accounts based on numbers of followers.

## Discussion

In this paper we address two research questions at the core of the digital misinformation problem. Specifically, we compare the efficacy of several metrics in identifying superspreaders of low-credibility content on Twitter (RQ1). We then employ the best performing metrics to qualitatively describe these problematic accounts (RQ2).

The *h*-index and Influence metrics display similar (and near-optimal) performance in identifying superspreaders. However, the accounts identified by Influence share a larger proportion of tweets that do *not* link to low-credibility sources. This makes the *h*-index preferable as a tool to identify superspreaders of low-credibility content because mitigation measures are likely to remove or restrict the spread of *all* information shared by those accounts. On the other hand, some bad actors may intentionally post harmless content to mask their deleterious behavior.

The dismantling analysis reveals a striking concentration of influence. It shows that just 10 superspreaders (0.003% of accounts) were responsible for originating over 34% of the low-credibility content in the eight months following their identification. Furthermore, a mere 0.25% of accounts (1,000 in total) accounted for more than 70% of such content. This highlights the significant role of these superspreaders, further exacerbated by their use of more toxic language than that of average content sharers.

A manual classification of the active superspreaders we identify reveals that over half are heavily involved in political conversation. Although the vast majority are conservative, they include the official accounts of both the Democratic and Republican parties. Additionally, we find a substantial portion of nano-influencer accounts, prominent broadcast television show hosts, contrarian scientists, and anti-vaxxers. While past research has identified influencers responsible for the spread of fake news within political discussions [34, 35], few of our identified superspreaders overlap with those found in this work. This discrepancy is likely due to our broader focus on general low-credibility content. Moreover, we employ distinct methodologies for categorizing content as low-credibility and pinpointing influential accounts. These methodological differences enable us to uncover a wider array of actors who illustrate various motivations for spreading untrustworthy content: fame, money, and political power.

Our analysis shows that removing superspreaders from the platform results in a large reduction of unreliable information. However, the potential for suspensions to reduce harm may conflict with freedom of speech values [55]. The effectiveness of other approaches to moderation should be evaluated by researchers and industry practitioners [56]. For instance, platforms could be redesigned to incentivize the sharing of trustworthy content [57].

The current work is specifically focused on original posters of low-credibility content and their disproportionate impact. However, it opens the door for future research to delve into the roles of "amplifier" accounts that may reshare misinformation originally posted by others [8].

This study relies on data obtained prior to Twitter's transformation into X. At that time, Twitter was actively experimenting with ways to mitigate the spread of misinformation [58]. This is starkly contrasted by X's recent decisions to lay off much of their content moderation staff and disband their election integrity team [59, 60]. Despite these changes, the key mechanism studied here—a user's ability to reshare content—remains a fundamental aspect of the platform.

Internal Facebook documents detailed a program that exempted high-profile users from some or all of its rules [61]. Evidence presented in this paper suggests that Twitter was also more lenient with superspreaders who were verified or had large followings. Social media platforms may be reluctant to suspend prominent superspreaders due to potential negative publicity and political pressure. Paradoxically, the more prominent a superspreader is, the greater their negative impact, and the more difficult they are to mitigate.

## Acknowledgments

The authors thank Drs. Yong-Yeol Ahn and Alessandro Flammini for their helpful discussions about this project. We also thank the members of the Networks & agents Network (NaN) research group at Indiana University, for feedback on preliminary results, as well as Dr. Yonatan Zunger for helpful input related to the tested metrics.

## Author Contributions

**Conceptualization:** Matthew R. DeVerna, Rachith Aiyappa, Diogo Pacheco, John Bryden, Filippo Menczer.

**Data curation:** Matthew R. DeVerna, Rachith Aiyappa, Filippo Menczer.

**Formal analysis:** Matthew R. DeVerna, Rachith Aiyappa.

**Funding acquisition:** Filippo Menczer.

**Investigation:** Matthew R. DeVerna, Rachith Aiyappa.

**Methodology:** Matthew R. DeVerna, Rachith Aiyappa, Diogo Pacheco, John Bryden, Filippo Menczer.

**Project administration:** Matthew R. DeVerna, Rachith Aiyappa.

**Resources:** Filippo Menczer.

**Software:** Matthew R. DeVerna, Rachith Aiyappa, Filippo Menczer.

**Supervision:** John Bryden, Filippo Menczer.

**Visualization:** Matthew R. DeVerna.

**Writing – original draft:** Matthew R. DeVerna, Rachith Aiyappa, Diogo Pacheco, John Bryden, Filippo Menczer.

**Writing – review & editing:** Matthew R. DeVerna, Rachith Aiyappa, Diogo Pacheco, John Bryden, Filippo Menczer.

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
