## [Decision Letter · Decision Letter 0]

13 Dec 2023

PONE-D-23-36459Identification and characterization of misinformation superspreaders on TwitterPLOS ONE

Dear Dr. DeVerna,

Thank you for submitting your manuscript to PLOS ONE. After careful consideration, we feel that it has merit but does not fully meet PLOS ONE’s publication criteria as it currently stands. Therefore, we invite you to submit a revised version of the manuscript that addresses the points raised during the review process. Some of the concerns raised by Reviewer #1 are especially relevant.First, you should consider reframing your paper, at least in part. The use of low credibility domains to identify misinformation and of high volume of retweets as a measure of spreading power are both common habits in the related literature, yet both questionable. In particular, the term "superspreader" has a precise meaning in epidemiology which is not the one it has in your paper.Second, you should be more explicit about some of your methodological choices. To the best of my understanding, none of the evaluation metrics is computed on the (partially) dismantled network, hence the rationale of using a dismantling procedure is unclear. Also, you did not provide a mathematical formula for your Influence metrics, which prevents the reader from understanding the difference between Influence and the numerator of the M_i evaluation metrics.

Finally, I agree with the Reviewer that renaming the h-index is not a good idea.

We look forward to receiving your revised manuscript.

Kind regards,

Stefano Guarino, Ph.D.

Academic Editor

PLOS ONE

Journal Requirements:

4. Ethics statement only appears at the end of the manuscript:

Your ethics statement should only appear in the Methods section of your manuscript. If your ethics statement is written in any section besides the Methods, please move it to the Methods section and delete it from any other section. Please ensure that your ethics statement is included in your manuscript, as the ethics statement entered into the online submission form will not be published alongside your manuscript.

Reviewers' comments:

Reviewer's Responses to Questions

**Comments to the Author**

1. Is the manuscript technically sound, and do the data support the conclusions?

Reviewer #1: Partly

Reviewer #2: Yes

2. Has the statistical analysis been performed appropriately and rigorously? 

Reviewer #1: Yes

Reviewer #2: Yes

3. Have the authors made all data underlying the findings in their manuscript fully available?

Reviewer #1: No

Reviewer #2: Yes

4. Is the manuscript presented in an intelligible fashion and written in standard English?

Reviewer #1: Yes

Reviewer #2: Yes

5. Review Comments to the Author

Reviewer #1: Overview

Every review should open with a bit of praise, and there’s plenty to work on for dolling that out here. The authors have compiled a unique data, one that we probably won’t see recreated again anytime in the next several years. They work through a careful and quantitative analysis, asking clear questions and explaining both their goals and methods in a clear manner. This paper should be published in PLoS ONE and I hope it is. However, before I can accept the manuscript I think it needs a pretty serious makeover in terms of its framing. Not only do I think this will make the paper more “correct”, I think it will help make it a more valuable piece because it will provide deeper insight into a specific phenomenon and not get washed away in the flurry of pieces about superspreaders and misinformation. I’ll review that concern in the overview, my thinking behind it, and provide some examples of why I believe the current framing may be misleading. By all means, the authors should feel free to push back but hopefully they’ll take this to heart. I get that there’s a lot below, but the quantity is a reflection of what I think the paper can do, more than what it didn’t do!

The authors operationalize misinformation superspreaders based on various network features, their propensity to share misleading content, and how far the misleading content they share/amplify tends to travel. While low-quality sources are likely to be misleading, I don’t think this really encapsulates the variety of misinformation shared during the pandemic and the strategies by which it was spread.

For instance, an article in the BMJ amping up an ostensible Pfizer whistleblower was the most widely shared scientific article in the pandemic with 78330 shares via altmetrics. Interestingly, the originator of this intentionally misleading piece, Paul Thacker, is cited in the present manuscript as ref 46. Thacker has been responsible for quite a bit of anti-vaccine misinformation, either as a source or resharing.

More generally, scientific articles were often weaponized, as were text-based threads, images, and videos shared on platform. The University of Washintgon’s IHME models fit pandemic dynamics to a gaussian function and led to predictions that the “wave would pass” and the pandemic would be over within months. Several faculty at Stanford early on produced academic research arguing that no more than 10,000 Americans would die and stood by that claim even when it was shown unequivocally to be false by some of the worlds’ top statisticians. Similarly VAERS data, reports of random death, etc… was weaponized repeatedly by anti-vaccine communities to create misleading narratives about vaccines. Even the WHO maintained that covid was not airborne for two years, against scientific consensus. This isn’t peculiar to pandemic-related misinformation, but also applied heavily during the election as well. Some accounts simply used text and made misleading statements, others relied on images, links to content on youtube (this was big), etc… and would not be swept up in a definition based on sites ranked low in quality.

There’s no reason to believe that misinformation super-spreaders all use the same strategy, particularly across the political spectrum or various ideological divides that emerged during covid. My first concern is that the write up is about super-spreaders of misinformation but it’s really getting at super-sharers-of-low-quality-websites. I think the paper should be reframed to perhaps frame this as super-spreaders of low-quality information rather than misinformation.

A related issue is that, as far as I can tell, retweets are linked to the original source, not to the person who caused the retweet to be in an individuals’ feed. I think this is a bit of an issue, because often super-spreaders amplify misinformation rather than generate it. The EIP report details this phenomenon at length. In disease, superspreaders would typically not refer just to index cases, as here, but to anyone whose individual reproductive number is high (i.e., anyone in the transmission chain who forwards it along to many others). As such, I worry that the superspreader definition in the sense of Lloyd-smith 2005 might be quite different from how its described here. There should be some discussion clarifying this difference and the write-up should caution that these are influential original sources of low-quality link sharing rather than super-spreaders of misinformation.

The reason this is more than pedantic is that some of the most influential accounts wield that power via retweets of smaller sized accounts. Our understanding of super-spreading here actually presents quite a bit of conflict with the operationalization in the study. We know, from empirical work, theory and simulations that large accounts that retweet are responsible for a substantial amount of spread and their influence here gets foisted upon the person they tend to retweet. If you were tasked with removing an account, basing it on the definition in the paper would have a smaller impact on the spread of misinformation than if we had information about account amplification. For example, let’s say there’s an account that is smaller and repeatedly posts misinformation that is amplified reliably by an account that follows them with 100M followers. FIB-index and influence here would suggest removing the smaller account would reduce spread of misleading links. Upon doing so, the 100M follower account simply amplifies a different account also sharing misleading links.

I think all of the above are very real and insurmountable changes for framing this paper as a study of super-spreading of misinformation. That said, they present no problems to a study on influential originators of information from low-quality websites.

The main thing I would like to see in the rewrite is to be a bit more cautious about definitions and write this up as what it is---a clear study on the influential originators of widely shared links to misleading websites. I think that is quite different from super-spreaders of misinformation, but nonetheless an interesting and important contribution to the literature. I understand this is less catchy, but I think the study has either the option of being a fairly flawed study of superspreading of misinformation or a very good study who influentially shares links to low-quality websites. I am sure there is overlap in these populations, but I worry that we’re missing most of the iceberg here.

https://www.bmj.com/content/375/bmj.n2635

Additional more serious points

The dismantling analysis is interesting, and it’s clear that it demonstrates a consistency between the observation and evaluation period (i.e., people who share sources from low quality sites tend to continue to do so). At the same time, it’s not clear the extent to which the relative performance of the metric merely reflects tautology between the definition of the retweet network and the definition of the metric. While I understand the analysis, it’s unclear what it’s telling us beyond that the retweet network is pretty stable, and that h-index and Influential both measure retweets within the retweet network. Perhaps there’s a less convoluted way to show this?

For the political leaning bit, I think this is somewhere that the difference between misinformation and source quality really kicks in. Several very prominent left-leaning accounts shared substantial misinformation about the pandemic, but often made their case by exaggerating news from reliable sites. As terrible as the pandemic was, it wasn’t--for instance--airborne HIV or various other narratives that took off.

Throughout: Estimates need bounds placed on their uncertainty (e.g., confidence intervals, credible regions etc…) wherever possible. Somewhat relatedly, I’d be careful about inferences drawn from the presence/absence of statistical significance in various places. Ideally anywhere that has an estimate or a p-value should have an interpretable effect size (e.g., not just a test statistic like U) with quantified uncertainty.

Minor Points:

L11: The second paragraph of the introduction is just a single sentence, consider folding it into P1 or P2.

I urge you just to call the FIB-index the h-index. No point in renaming it again if it’s the same metric! Other research relying on it outside of its original content typically uses the same term and this will be more searchable/clearer to the reader.

Reviewer #2: The manuscript "Identification and characterization of misinformation superspreaders on Twitter" by Matthew R. DeVerna et al. presents a study of misinformation superspreaders on Twitter and proposes a method to predict which accounts will become superspreaders. The work is interesting and relevant. The methodology is sound. I have only minors points that the authors can easily address. Apart from this, I recommend its acceptance.

The main point to address is the fact that Twitter does not really exists anymore, or at least not in the same form than it was during this study. I think the proposed method can be generalized to other platforms, so this is not a crucial issue, but the authors only used Twitter data, so they should still address this point in the introduction and discussion.

The second point is that I think the authors could refer to the results of two other studies that have looked closely at Twitter misinformation superspreaders:

- Bovet, A., & Makse, H. A. (2019). Influence of fake news in Twitter during the 2016 US presidential election. Nature Communications, 10(1), 7. https://doi.org/10.1038/s41467-018-07761-2

- Flamino, J., Galeazzi, A., Feldman, S., Macy, M. W., Cross, B., Zhou, Z., Serafino, M., Bovet, A., Makse, H. A., & Szymanski, B. K. (2023). Political polarization of news media and influencers on Twitter in the 2016 and 2020 US presidential elections. Nature Human Behaviour, 7(6), 904–916. https://doi.org/10.1038/s41562-023-01550-8

These studies report the fraction of verified/unverified/deleted, politically or media affiliated top misinformation superspreaders in 2016 and 2020, so it seems very natural to summarize them in the related work and compare what the authors found with them.

The third point is about the account classification (Fig. 2). 128 acounts were classified as not active anymore, but the authors should still look at what kind of accounts they were and included them in the other categories (media, political, ...).

Another point that the authors should comment on is how the fact that Twitter was actively combating misinformation in 2020 could impact their result.

A small detail: it's not clear in the text of r_m is the "ratio of tweets linking to low-credibility content" (line 239) or if it is the "proportion of shared sources linking to low-credibility content" (line 252), i.e. if the same source is counted multiple times or not.

6. PLOS authors have the option to publish the peer review history of their article (what does this mean?). If published, this will include your full peer review and any attached files.

Reviewer #1: No

Reviewer #2: No

---

## [Author Response · Author response to Decision Letter 0]

31 Jan 2024

Please see the "Response to Reviewers" PDF document for a detailed response.

---

## [Decision Letter · Decision Letter 1]

1 Apr 2024

Identifying and characterizing superspreaders of low-credibility content on Twitter

PONE-D-23-36459R1

Dear Dr. DeVerna,

We’re pleased to inform you that your manuscript has been judged scientifically suitable for publication and will be formally accepted for publication once it meets all outstanding technical requirements.

Kind regards,

Stefano Guarino, Ph.D.

Academic Editor

PLOS ONE

Additional Editor Comments (optional):

Dear Authors,

One of the two original reviewers didn't submit their review to the revised version of your paper, after having accepted to do so. I tried finding a second reviewer willing to take on the assignment, but to no luck. Finally, I decided to review the paper myself, especially paying attention to whether the comments by that reviewer had been addressed thoroughly, and I believe you did a fairly good job. I am sorry if this slowed down the whole process.

Reviewers' comments:

Reviewer's Responses to Questions

**Comments to the Author**

1. If the authors have adequately addressed your comments raised in a previous round of review and you feel that this manuscript is now acceptable for publication, you may indicate that here to bypass the “Comments to the Author” section, enter your conflict of interest statement in the “Confidential to Editor” section, and submit your "Accept" recommendation.

Reviewer #2: All comments have been addressed

2. Is the manuscript technically sound, and do the data support the conclusions?

Reviewer #2: Yes

3. Has the statistical analysis been performed appropriately and rigorously? 

Reviewer #2: Yes

4. Have the authors made all data underlying the findings in their manuscript fully available?

Reviewer #2: No

5. Is the manuscript presented in an intelligible fashion and written in standard English?

Reviewer #2: Yes

6. Review Comments to the Author

Reviewer #2: I thank the authors for addressing my issues.

This is not crucial, but I would have appreciated if the authors would have done a better comparison of their results with the related works refs [34,35]. In particular, the approach to find superspreader is very similar: network percolation and dismantling are the same thing. Although, I agree that the implementations and the goals of the analysis are different. They could have also compared the top superspreaders they found with the superspreaders found in these papers.

Another point is the data availability. The authors share the tweet IDs, which was the common practice when the Twitter API was still available for researchers. But this is not the case anymore with X. This is unfortunate and the authors are not to blame for this at all. But I think they should update their data availability statement accordingly.

7. PLOS authors have the option to publish the peer review history of their article (what does this mean?). If published, this will include your full peer review and any attached files.

Reviewer #2: No

---

## [Editor Report · Acceptance letter]

26 Apr 2024

PONE-D-23-36459R1 

PLOS ONE

Dear Dr. DeVerna, 

I'm pleased to inform you that your manuscript has been deemed suitable for publication in PLOS ONE. Congratulations! Your manuscript is now being handed over to our production team.

Kind regards, 

on behalf of

Dr. Stefano Guarino 

Academic Editor

PLOS ONE